# European Rural Demographic Strategies: Foreshadowing Post-Lisbon Rural Development Policy?

**Thomas Dax** [1,*] and **Andrew Copus** [2]

1   Federal Institute of Agricultural Economics, Rural and Mountain Research, 1030 Vienna, Austria
2   Faculty of Social Sciences and Business Studies, Karelian Institute, University of Eastern Finland, 80100 Joensuu, Finland
*   Correspondence: thomas.dax@bab.gv.at

**Abstract:** The European Commission's Long-term Vision for Rural Areas, published in June 2021 and building on a previous report on the Impact of Demographic Change and a Green Paper on Ageing, underlines the importance of population trends as a key issue for EU rural policy. The increasing concern about demographic issues, especially in rural Europe, has been accompanied, and in some cases preceded, by the publication of national population strategies. This renewed interest within the European policy community probably has roots in politics rather than new research or fresh evidence. Rural depopulation is not a new phenomenon, nor is it a new research topic. Nevertheless, to better understand this renewed interest, it is instructive to review recent scholarship and consider whether there is any evidence that the processes and systems of rural and regional (demographic) development are delivering new kinds of challenges, requiring refreshed policy approaches. Having established this context, we present a comparative review of a selection of national strategies, identifying shifting perspectives on goals, the instruments proposed, and implied intervention logics. Arguably, cumulative evidence points to an incremental shift of the policy discourse away from neoliberal, Lisbon-inspired visions of rural competitiveness and cost-effectiveness and towards a quest for rural well-being, rights to basic services, and more (spatially) inclusive rural development. This increasing emphasis on qualitative change may be symptomatic of a wider shift in the zeitgeist of rural policy, reflecting a number of globalised trends, including an awareness of the potentials and limitations associated with changing patterns of inter and intra-regional mobility.

**Keywords:** shrinking regions; rural development; rural policy; demographic trends; local development; intervention logic; participation; mobility changes; place-based policies; territorial attractiveness

## 1. Introduction

The changing role of population trends in rural development reflects both the evolving macro-demographic and mobility context and a shifting "zeitgeist" in terms of policy paradigms [1]. In this paper, we present observations about recent national population strategies as evidence of a shift away from neo-liberal growth objectives and assumptions, measured in economic terms, towards wider and more qualitative aspirations for well-being and geographic inclusion, in the face of seemingly ineluctable demographic trends.

Population decline (shrinking) has been an observed phenomenon in many parts of rural Europe for as much as two centuries, and the history of interventions to slow or reverse it is almost as long. Until the end of the last century, in the context of a simplistic binary view of space as either urban or rural, "urbanisation" tended to be implicitly assumed to be a necessary sacrifice in exchange for economic development. Under this premise, rural development was primarily understood as an endless quest to alleviate the ever-increasing performance gap between urban and rural economies. As out-migration and human capital depletion from rural regions were perceived as the inevitable result, combatting population decline became a prime development objective for rural regions.

More recently, counter-urbanisation and the recognition of dynamic and prosperous rural areas have changed the perception of "shrinking" areas to that of a distinct group within a diverse array of types of rural environments. Within the context of EU rural policy, eastward enlargement, the medium-term impacts of the 2008 financial crisis, and the subsequent rise of rural populism have pushed the issue of rural shrinking up the policy agenda. Meanwhile, the economic, social, and spatial processes commonly driving shrinking have shifted, and rural policy paradigms have evolved, so that the renewed interest, exemplified by the Commission's Long-term Vision for Rural Areas, together with a number of national initiatives, cannot be characterised as "more of the same", or a simple redoubling of efforts. This paper explores this shifting nexus of context and policy and reflects upon the possibility that evolving demographic strategies have signalled a subtle shift in the overarching goals of European (EU and national) rural development policy.

### 1.1. EU Policy, from Less Favoured Areas to Smart Villages

To set the context, and to aid the interpretation of the direction in which current EU policy seems to be heading, a brief account of its past evolution will be helpful. Many aspects of EU policy have direct or indirect effects on rural population trends [2]. Here, we will focus on (arguably) the most important; rural development, cohesion policy, and community-led local development (CLLD, including LEADER).

Within the complex evolution of EU rural development policy, two distinct phases in terms of the response to rural shrinking may be identified [3]. In the first phase, beginning in the 1970s and fading out in the middle of the first decade of this century, rural depopulation was addressed by exogenous and sector-focused compensation for geographical disadvantage, exemplified by less favoured area (LFA) policy. The second stage emerged in the closing years of the last century with the Agenda 2000 Reform and the establishment of CAP Pillar 2. This evolved as a menu of measures which, driven by Lisbon employment and economic growth priorities (rather than the need to retain rural population), became increasingly sectoral rather than territorial in implementation. The concept of multi-functionality justified substantial support for agri-environmental measures, while measures to support the wider rural economy (and population) received a relatively small share of funding in most member states.

The history of cohesion policy's approach to rural depopulation is distinct and increasingly divergent from that of rural development policy. During a brief period in the 1990s, European Regional Development Fund (ERDF) programmes contributed directly to population retention through integrated (multi-fund) and spatially targeted programmes (Objective 1, 5b, and 6). However, since the abandonment of multi-fund programming and replacement of Objective 5b programmes by Rural Development Programmes (RDPs) devised and managed by Member States, Cohesion Policy has principally manifested itself in city-anchored innovation/growth paradigms in which benefits to rural areas depend heavily upon assumed "spread effects" [4]. This was accompanied by an ongoing discussion, by the EC in collaboration with the OECD, of urban–rural relationships and advocacy for greater spatial cooperation [5,6].

As a "Community Initiative", LEADER, in its various reincarnations, including, in the current programming period, as a manifestation of community-led local development (CLLD), sits alongside rural development and cohesion policy, distinctive in its emphasis upon "bottom-up" diagnosis and prioritisation of each local action group (LAG) area's needs. As such, it exhibits a wide range of responses to population decline, and a variety of intervention logics. In principle, such (neo-) endogenous approaches are more likely to be effective in meeting local needs, though much depends upon available evidence, priorities, and the quality of decision-making. These reflect not only local institutional capacity and human capital, but also the level and nature of support from national capitals and Brussels.

EU policy is typically characterised by incremental change and evolution rather than radical restructuring, and this has been the case in terms of the recent upsurge of interest in tackling rural demographic challenges. The initial signs of this were a series of reports,

commissioned by the European Parliament (EP) [7–9], the Committee of the Regions [10,11], and the Economic and Social Committee [12]. It was also reflected in the establishment of an EP Intergroup on Rural, Mountainous, and Remote Areas (RUMRA) and the appointment of a commission vice president with a specific responsibility for "Democracy and Demography". This was followed by a commission report on the impact of demographic change [13], a green paper on ageing [14], and, after an extensive consultative process, the Commission's "Long Term Vision for Rural Areas" [2,15]. It is important to be clear that this process has not resulted in new policy structures (such as the creation of a new instrument or fund), but it has raised the priority of responding to demographic trends across a broad range of EU policy instruments and strategies.

In the absence of any significant structural changes, attention must turn to available tools or approaches, which may be developed to address the rising concern. The most likely candidate would seem to be the concept of "smart villages", which has been promoted by the European Network for Rural Development (ENRD). Arguably [3], the smart village initiative is a repackaging of neo-endogenous development principles, sometimes (but not necessarily) incorporating new technology and digitalisation. The use of the word "smart" is indicative of conceptual legacies from both "smart specialisation" [16,17] and "smart shrinking" [18–20]. It is important to be clear that smart village approaches may be formulated to address a range of long-standing rural development concerns (geographic disadvantages, remoteness, sparsity, peripheralisation, inner peripheries, or capital deficiencies in various forms), which are not necessarily linked to depopulation. However, where they are a response to population decline, such as smart shrinking, they usually focus on adaptation and sustaining well-being rather than setting out to reverse the demographic trend.

### 1.2. Framing the Basic Concepts of Rural Shrinking

As a foundation for the discussion of policy responses, which is the heart of this paper, it will be helpful to briefly specify the concepts and terminology that frame the issue of rural demographic shrinking. Firstly, we would follow Grasland in confining our attention to rural areas in which negative demographic trends are established over a period "greater than or equal to one generation" [21] (p. 25). Temporary fluctuations triggered by ephemeral events are not the same as embedded and self-perpetuating downward spirals of long duration.

Secondly, it is worth distinguishing "simple" (demographic) shrinking, evidenced in terms of population counts, from the wider socio-economic process sometimes termed "complex" shrinking [22]. The latter, of course, cannot be ignored in any attempt to understand the drivers of demographic trends.

A basic distinction, which has obvious implications in terms of designing policy responses, is between "active" shrinking, in which negative net migration is the key driver, and "legacy" shrinking, in which an age structure affected by past migration processes results in high mortality rates, low fertility rates, and negative natural demographic change. In twenty-first-century rural Europe, the latter has become more widespread, with active shrinking processes being more typical of eastern and southern member states, though even here they are superimposed on legacy effects [23]. Legacy effects are generally manifest in age-specific population trends, the shares of working-age and fertile populations, and dependency rates.

### 1.3. What Recent Demographic Research Tells Us about Rural Shrinking

The "grey literature", which documents the increasing interest in rural demography as a European policy concern, makes very little reference to recent theoretical debates in the fields of demography, rural geography, economics, or sociology. Equally, the theoretical literature on demography tends to have a national or international focus, and regional or rural analysis is much less common [24]. Nevertheless, recent demographic research provides some very interesting and important insights. These relate to two broad themes, which are

presented below. The first reflects the perspective that observed rural demographic trends, and migration behaviour in particular, are not necessarily part of a constant, unchanging spatial process but instead are subject to either medium-term cycles or long-term shifts. The second theme focuses on the characteristics and behavioural motivations of individuals and households, not only those that migrate, but also those that prefer to stay in rural areas. This research has important implications for our understanding of rural trends.

### 1.3.1. Migration Processes Are Not Constant through Time

The demographic transition is a long-established theory [25,26] that describes the way in which most societies pass through a sequence of stages, beginning with (pre-industrial) high fertility and mortality rates and transitioning towards an industrialised and urban stage, characterised by both lower fertility and mortality rates. In the early stages, population growth occurs because of the lag between the trends in fertility and mortality. As originally formulated in the 1920s, the final stage was a steady state, a new low-level equilibrium. Some recent literature [27–29] raises the possibility that the final stage, or end point, might not be a new equilibrium but a sustained decline. If this is the case, then clearly a mitigation policy for shrinking rural areas is very challenging, to say the least. The literature relating to the demographic transition says little about the implications for rural areas. However, Coleman and Rowthorn [28] (p. 219) express a dismissive neo-liberal view: "Unless accompanied by concomitant decline at the national level, the internal shifting balance is devoid of strategic or major political consequences and should generally be regarded as a natural and desirable consequence of adjustment to changed realities of comparative regional advantage".

The second strand of literature is more empirical and inductive in approach. It presents evidence for a gradual reduction in migration, first observed in the USA by Cooke [30] in the closing decade of the twentieth century. Since then, careful analysis has suggested that this may be a societal shift, independent of the effects of economic cycles and legacy age structure effects [31–34]. This increasing reluctance to relocate, memorably captured by Cooke's phrase "secular rootedness" has been explained in a variety of ways, social, economic, and technological. Whatever the answer about the specific drivers, the implications for mitigation approaches for shrinking rural areas are clear, retention of existing residents is likely to be more successful than efforts to attract in-migrants.

A third strand of research is exemplified by the detailed analysis of the effects of boom and bust on the redistribution of population between rural and urban areas in Greece by Salvati [35]. Migration flows (mostly out of rural areas) were much stronger during times of economic growth than in times of economic crisis. This suggests that rural population policies need to take careful account of cycles in the national economy.

On the one hand, the foregoing evidence that inter/intra-regional migration may not be relied upon as a source of mitigation for rural shrinking tends to be "drowned out" by the narrative of increasing international flows, especially asylum seeking and economic migration. On the other hand, academic interest in multi-local and transnational living arrangements [36,37] reminds us of the increasing complexity of the twenty-first century context.

### 1.3.2. Understanding Individual Migration Behaviour and Motives

As mentioned above, there is some evidence from a variety of national contexts that levels of (internal) migration activity may be lower in the future than they have been in the past. This raises questions about the degree to which rural shrinking mitigation initiatives should rely on "place making" to attract in-migrants. Perhaps greater attention should instead be paid to the retention of existing residents, especially the young, well-educated, and enterprising individuals who, during periods of urbanisation, were the first to leave. The characteristics and motives of "stayers", especially "life-stage" effects, have been explored by a number of researchers in recent years [38,39]. Cultural attitudes to "staying"

(especially if held by policy practitioners) would appear to be a very significant factor affecting the likelihood of success for mitigation initiatives:

> *"Understanding the staying processes of these young people can lead to improved policy interventions to help stem the ongoing rural brain drain and ageing of rural populations. Inadvertently, policymakers may view staying from a mobility perspective: Stayers have "failed to leave" or "been left behind." Young adults are not always recognized as a rural asset that should be retained. Yet many are highly educated, possess a strong sense of belonging based on familial ties that go back several generations, and exert considerable human agency to stay. Such young people are arguably the lifeline of many rural communities"* [38] (p. 9–10).

"Life stage" perspectives on migration patterns in different national contexts illustrate the variety of pathways that influence demographic trends. In Northern Australia, in-migration of the young and economically active is not problematic due to workforce demands from resource-based industries [40]. On the other hand, retention of the same individuals when they reach middle age or retirement is much more difficult. This is a mirror image of life-stage effects in Northern Europe and serves as a reminder that tailored solutions are necessary.

## 2. Conceptualising the Policy Response

Policy responses to rural shrinking may be differentiated on the basis of both *what* they aim to achieve and *how*, in practice, they go about pursuing their objectives.

### 2.1. Mitigation and Adaptation

As in the realm of climate change, two broad responses to shrinking may be distinguished: *mitigation* (changing the trend) and *adaptation* (neutralising the effects of decline). The first of these is much easier to "sell" in a political context. Elections are difficult to win on a platform of adapting to inevitable decline [41,42] and, in consequence, strategy documents tend to present visions of restored growth. However, since legacy effects are "baked into" age structures, effective mitigation is essentially about attracting in-migrants, and discouraging out-migration. Both are very challenging, especially in the context of sensitivities regarding international migration. Adaptation, on the other hand, is often, for pragmatic and implementation reasons, the dominant response for local or regional actors. Where these actors have responsibility for the delivery of services (education, health, waste collection, utilities, etc.), the objective is usually to "balance the books" within the context of rising unit costs exacerbated by increasing sparsity. Thus, if we consider the overall response of local government in terms of "net effects" on population trends, it is often a hybrid of well-publicized support for growth alongside a range of low-profile adjustments to services that effectively prioritise adaptation.

Closely related to the choice of mitigation or adaptation approaches is the ultimate goal of intervention, whether explicitly articulated or implied. The conventional thinking in rural development circles has always been that "growth is good" and that depopulation is something that prompts corrective interventions. Local politicians are generally very sensitive to the fact that there are few votes to be gained by accepting the inevitability of their constituency's demographic trajectory [41,42]. The majority of policy documents at the local, regional, and even national levels continue to set population growth as a goal, even if there is strong evidence to suggest that this is unrealistic. However, a number of euphemisms for adaptation are beginning to creep in, especially in international contexts, where the pressure of democratic consequences is less direct. Thus, the European Commission's Long-Term Vision for Rural Areas [2] talks of "balanced territorial development", whilst the OECD's Rural Policy 3.0 [43] talks frankly of "adapting to demographic change" and shifting the focus to economic growth and well-being. There is some evidence [44] that the populations of more crowded countries, such as the Netherlands, accept rural shrinking as inevitable, or even welcome it.

From an academic perspective, a similarly pragmatic view is expressed very clearly by Pinilla and Saez [45], who argue against "defining fascinating goals which are impossible to achieve, as if we were able to change structural and global trends" (p. 341). For sparsely populated areas, they argue, "the relevant question is not how many people live there. In a digital and global world, thresholds are changing every day. The question is whether people are able to live in a remote rural area because they want to live there. A better life in remote rural areas is the objective, how many is the consequence" (pp. 344–345).

### 2.2. A Simplified Theory of Change Perspective

It would not be practicable or perhaps even helpful to implement a full theory of change (ToC) analysis [46–54] here. In any case, the national policy documents to which we shall shortly turn do not provide sufficient detail to allow this. Nevertheless, the underlying perspective of ToC, which emphasises the interaction between drivers and contextual conditions on the one hand and strategies, interventions, intermediate outcomes, and goals on the other, is extremely helpful. We therefore believe that paying attention to key elements of the approach can provide a helpful interpretation framework for assessing strategic choices embedded in policy responses to rural demographic trends.

Our simplified ToC approach will seek to highlight two key features of the national strategies that, we believe, capture the (often implicit) perceptions and concepts which determine the style of intervention:

1. The first of these is the way in which the *diagnostic narratives* of rural depopulation are presented, and the analogies employed.
2. The second is the nature of the "*intermediate outcomes*", which are embedded in the strategies. These may be envisaged as "stepping-stones" on the way to addressing demographic decline. As such, they are symptomatic of a strategy's underpinning conception of complex shrinking, and of appropriate responses.

In the ESPON ESCAPE project [53] (p. 47) and [54] (p. 24ff), a ToC approach was the basis of a retrospective consideration of four generic approaches to shrinking: compensation, relocalisation, global reconnection, and smart shrinking. Since each of the national strategies described below contains elements of one or more of these approaches, it will be helpful to provide a very brief explanation here (Table 1).

**Table 1.** Key ToC Features of Four Generic Policy Approaches to Rural Shrinking *.

| Generic Approach | Compensation | Relocalisation | Global Reconnection | Smart Shrinking |
|---|---|---|---|---|
| **Diagnostic Narrative** | Population loss is due to the decline of traditional resource-based economic activities and the geographical disadvantages they face. | Economic activities in remote rural areas are at a disadvantage in the globalised economy. Long supply chains mean added value is retained in cities. | Rural business networks in remote areas are not well connected with the global economy. Resulting knowledge and innovation lags lead to stagnation and fewer employment opportunities. | Some degree of population decline is an unavoidable adjustment to a changed economic context. The response should be to manage that decline in such a way that minimises well-being impacts. |
| **Intermediate Outcomes** | Compensation and income support maintain the viability of economic activities, stemming outmigration. | Viability may be restored (and out-migration reduced) by shortening supply chains and closer ties to adjacent markets. | Improved access to globalised sources of knowledge and innovation enhances business performance, creates quality jobs, improves population retention, and attracts in-migrants. | A smaller economy but improved well-being for the residual population through innovative service delivery, often employing new IT-based solutions. |

* Source: Author's elaboration, based upon [53,54].

The presentation of recent national strategies for Spain, Italy, Germany, Scotland, and France that follows will seek to highlight the diagnostic narratives and intermediate outcomes that provide clues to the changing paradigms that they embody.

## 3. National Policy Initiatives in Rural Regions with Population Decline

In recent years, interest in strategies and policies to cope with rural regions long affected by population decline has intensified in many European countries and regions. Dedicated documents addressing this specific policy focus have been elaborated at different scales, reflecting a wide array of concerns and proposing a variety of responses that have been integrated through various approaches to policy coherence.

The current popularity of "shrinking rural regions" terminology might give the impression that this is a new focus area for national policies. However, it has long been a cornerstone for national policies with territorial cohesion objectives, directed at "disadvantaged", "marginal", or "peripheral" places. A conspicuous and well-publicised example of a long-established policy is the Italian Strategia nazionale per le aree interne (SNAI—National Policy for Inner Areas), which is now almost twenty years old. In Spain, Pinilla and Saez [45] (p. 339) liken the policy narrative to the curse of Sisyphus, with each new strategy taking the policy boulder back to the bottom of the implementation hill. Thus, we would argue that the origins of many national strategies predate those of the more recent European Commission renewal of interest in the issue. A comprehensive history of national policies would be an undertaking beyond the scope of this paper, and the examples provided below are presented as an illustrative selection of recent nation-wide strategies from major European countries (Spain, Italy, Germany, France, and the UK [Scotland]). The selected examples all have a focus on rural or regional population issues, as distinct from national-level strategies, which address national trends in the components of population change. The brief summaries that follow are intended to signpost the broad direction of travel in terms of (implicit or explicit) underpinning paradigm(s). The evolution of the discourse expressed by the different approaches reflects a shift from a primarily compensation diagnostic and policy understanding to more complex concepts of appreciating local assets, spatial interaction, and new perspectives on spatial organisation (see diagnostic narratives in Table 1). Such new perspectives might involve an array of diverse instruments with significantly different objectives, orientations, and effects on integration and involvement (see intermediate outcomes in Table 1). For a more detailed view on the five national strategies and specific examples of intervention measures, see profiles of national strategy documents in the Supplementary Materials.

### 3.1. Spain—National Strategy to the Population Challenge (2017)

Arguably, the country with the most intensive commitment to tackling shrinking rural regions is Spain. Public awareness of demographic challenges is high, and the political discourse is lively. While the national strategy document [55] was launched as recently as 2017, the large, sparsely populated areas of Spain have been a matter of concern for several decades. The popularity of the expression "Empty Spain" [56] and the creation of a Special Commission of the Senate to consider depopulation trends in mountain areas [57] are indicative of the quest for appropriate mitigation policies.

Policy urgency is driven by the ubiquity of demographic decline across rural Spain: 63% of all Spanish municipalities experienced population decline between 2001 and 2018. Almost 50% of these lost between 10% and 50% of their population during this period. Smaller municipalities are the worst affected; 90% of municipalities of less than 1000 inhabitants lost population [55]. The demographic process has been long established. Continuous out-migration since 1900 has created extensive sparsely populated areas, which have become "places of no relevance" [58]. These trends continue across "interior areas" of the country [59], particularly in mountain areas [57]. Positive demographic trends are found only in urban agglomerations and coastal areas. Out-migration of young people, resulting in demographic ageing and recursive effects on fertility rates, has been shown to

be key elements of the process, in south-west Spain [60], in the south [61] and Cantabria in the north [62].

In response to this bleak picture, the National Strategy is founded upon the core principle of "guaranteeing equal opportunities and free exercise of citizenship rights throughout the territory" [55] (p. 31). Equality of access to opportunities for men and women of all ages and in all places is the key objective (p. 4). Sustainability, well-being, access to basic services, territorial inclusion, and relationships between urban and rural areas are all emphasised. Rather than being the basis for a single programme or initiative, these goals are to be adopted across national government departments and also (reflecting federal governance structures) through regulations at the regional level. Consistent with this, the concept of "demographic proofing" is advocated.

Building upon and coordinating existing activity in depopulating areas certainly makes sense. Rising and widespread awareness of the complex bundle of drivers leading to a situation where there is seemingly no escape has resulted in numerous local initiatives spreading across the country. A dedicated national network [63] highlights positive local examples, including social projects for integrating various types of incomers, initiatives to raise the attractiveness of an area, to harness natural resources, to address mobility and digital shortcomings, to enhance community building, and to stimulate entrepreneurial innovation. In terms of the four generic approaches in Table 1, mitigation responses thus combine elements of relocalisation and global reconnection.

Many levels of administration have engaged in the current activities; in particular, regional administration launched processes to raise the attractiveness of remote places in order to cope with the problems of shrinking rural areas. These processes gained high-level recognition by serving as case studies for an OECD thematic project on rural regions with population decline [64]. The activities in the Macroregion Southwest Europe (RESOE) build on the regional strategies to mitigate population decline in the four Spanish autonomous communities: Asturias, Cantabria, Castilla y Leon, and Galicia.

### 3.2. Italy—Manifesto to "Repopulate" Remote Regions (2020)

In Italy, there has been longstanding awareness of the issues facing lagging regions and the challenges of remoteness due to the development gaps of the "Mezzogiorno" and the large share of mountain areas across the whole country. Population decline in mountain regions was the focus of one of the first nation-wide studies of the National Institute for Agricultural Economics (INEA) in the 1920s and 1930s [65]. Rising concern with ineffective instruments and limited policy outcomes to mitigate marginalisation processes led to the elaboration of a national scheme, the National Strategy for Inner Areas (Strategia Nazionale per le Aree Interne, SNAI), in 2012. SNAI provides local development efforts through national support for 72 small-scale regions of various types of remote areas, comprising 3.5% of the population and 16.7% of the total area of Italy [66]. As this programme constitutes complementary local support to the LEADER/CLLD approach, it underscores the national commitment for these areas. Although the challenge of depopulation is not explicit in the title of the initiative, these local activities address small-scale regions with on average 29,000 inhabitants and significant population decline (−4.4% 2001–2011). One of its main differences from the EU Cohesion Fund programmes is its firm anchoring in local participation and desire to nurture processes of local empowerment [67].

The discourse involves a wide range of disciplines, institutions, stakeholders, and local actors, revealing its pertinence throughout the country. The need to "escape from the trap of marginalisation" [68] has been widely discussed and is increasingly the consensus view. From the anthropological perspective, the firm commitment to listen to local inhabitants and attribute action to the local scale is underpinned [69]. Others underline the core objective of contributing to a "caring community", supporting strategies of well-being vs. technological and growth aspects [70]. Positive initiatives are spreading throughout Italian remote and mountain areas, leading to numerous examples of in-migrants engaging deliberately in settling in remote and mountain areas [71].

The SNAI and increasing concern for coping with areas left behind led to momentum for a national discourse on reverting the narratives on remoteness, marginalisation, and population decline, inspiring an expert-based initiative to "repopulate the remote parts of Italy" (Riabitare ltalia). The nation-wide discussion was based on a comprehensive analysis [72], a manifesto incorporating early experience of the COVID-19 pandemic in the first half of 2020 [73], and a spatial planning analysis focusing on the interrelations of spaces under the label of "Metromontagna" [74]. The initiative intends to stimulate creative contributions to narrative building and local action from actors at all levels through two research projects: The first focuses on places of community resilience within the COVID-19 pandemic. The second explores the creative power of young people establishing activities in remote places.

In terms of a diagnostic narrative, Riabitare ltalia is rich in ideas and metaphors, and it is not easy to identify a dominant analogy. However, two ideas stand out: The first (as in Spain) is that of the demographic *emptying* of localities (which implies the assumption of some kind of population capacity). The second is the need to reset the relationship between central and peripheral areas, not through market-based economic processes but through innovative social development and active citizenship and participation. Barca [75] (p. 551) summarises the objective as a heterodox view of remote places and a need for a radical turn. The Italian approach, particularly as expressed in Riabitare Italia, thus calls for strong relocalisation and global reconnection responses to the powerful forces of globalisation and metropolisation.

As in the Spanish strategy, Riabitare ltalia does not have a funded set of policy interventions and relies upon influencing the ongoing implementation of the SNAI, together with a range of other activities at national, regional, and local levels. As such, its "intermediate outcomes" are in the realm of changing hearts and minds, both in the policy sphere and within rural communities, rather than measurable economic impacts.

### 3.3. Germany—National Commitment to Cope with Inequality and Population Shrinkage (2020)

In July 2018, the German government established a commission on "Equal living conditions" to combat structural weakness and depopulation trends in rural and remote regions and to elaborate proposals for implementing concrete policy measures to achieve territorial justice. Through the strategy document "Our Plan for Germany—Equal living conditions everywhere" [76], the three responsible ministries provide a sound outline for addressing territorial disparities and enhancing action to curb depopulation trends. This recent strategic commitment was preceded by an intensive consultation exercise on coping with the demographic challenges facing almost all regions of Eastern Germany and increasingly rural parts of Western Germany too. Population declines in these parts of the country peaked after the reunification of Germany in the 1990s, being spurred both by out-migration and rapidly decreasing fertility rates. As a consequence, ageing trends in these rural regions are among the strongest of all European regions [77], and this has the effect of rapidly raising the role of migration in sustaining population levels.

One of the outcomes of the Living Conditions Commission's activity is the launch of a renewed regional development policy for structurally weak regions, which pools a set of 22 specific support programmes on enterprise support, innovation, skills enhancement, digitalisation, and infrastructure adjustment into a coordinated programme. The mid-term assessment of this programme reflects on the priorities of the strategy and grapples with the gap between the short-term focus on investment and growth, and the long-term transformation challenges of resource depletion, environmental degradation, and climate change [78]. Many regions, local actors, and experts contribute to the discussion of how creative use of existing and new policy instruments could finally provide a means to escape downward-spiralling trends. The focus here is on both novelty and practicability [79], as well as local engagement and intercommunal cooperation [80]. The long held negative views on the future for rural regions, particularly in Eastern Germany, are more and more replaced by "counterpoints" in narrative evolution that seek to find new constructions of

rural areas based on their place-specific assets, appreciation of uniqueness and diversity, and enhanced commitment of local actors [81]. As such, it aims at a combination of mitigation and adaptation strategies.

"Wrapping", the response to rural demographic trends in a living-conditions framework, is eloquent regarding the diagnostic narrative of the recursive relationship between well-being and demography, which underpins the German strategy. It is striking that, as in Spain and Italy, the discourse is no longer dominated by neo-liberal concepts such as competitiveness and urbanisation. Thus, in the broad sense, the intermediate outcome is that living conditions will be equalised across Germany, which (it is argued) will stem out-migration from and support attractiveness and in-migration to shrinking rural areas. At a more detailed level, the actions promoted include dispersal of civil service jobs, reducing variations in broadband speeds, improvements in rural train services, (debt reducing) support for municipal financing, enhanced childcare, and support for community engagement, volunteering, and social capital. It is not easy to characterise this response in terms of the four generic approaches in Table 1, although the emphasis on infrastructure points to global reconnection.

### 3.4. Scotland—A Scotland for the Future (2020)

The Scottish Government recently published a national demographic strategy: "A Scotland for the Future: The opportunities and challenges of Scotland's changing population" [82]. Like the Spanish, Italian, and German strategies, this document is partly aspirational, and partly an *ex ante* orchestration of many existing policy actions that—with hindsight—can be claimed to have beneficial side-effects in relation to the challenges and opportunities of the demographic trends.

However, unlike the examples above in the Scottish strategy, the terminology of "empty" rural areas is absent, being replaced by a diagnostic narrative based on the concept of (regional) "balance". "Balanced development" is perceived as characterised by some kind of "match" between population requirements and service provision capacity, housing, infrastructure, etc. There is also a labour market element—out-migration reduces the availability of skills and innovation capacity in areas experiencing shrinkage, while in "receiving" areas, service/housing/infrastructure demand exceeds capacity/provision.

Thus, a long-established drift of population from the rural west towards a more urbanised east is associated with; in the former, a pronounced tendency towards ageing, reduced fertility, and, in recent years, specific skill shortages; and in the latter, "overheating" of the housing market and pressure upon services. The clarity and simplicity of this characterisation reflect the strategy document's reliance upon statistics at the "Council Area" (local government) level, which mask a much more complex pattern of change at the local level [83]. It is also important to stress that the challenge faced by the west of Scotland is not negative net migration, but rather that current in-migration, despite exceeding out-migration, is insufficient and has the wrong age profile to offset the rate of natural decrease due to the legacy effects of out-migration in the past. The situation is further complicated by the ending of Free Movement following Brexit, which, reinforced by COVID-19 travel restrictions, has substantially reduced short-term migration of workers in the hospitality sector, agriculture, horticulture, and fishing, all of which are key economic drivers in the west of Scotland.

Despite the intrinsically "top-down" character of a national strategy, there is a strong thread of "place-based" implementation running through the Scottish document. Thus, the overall goal is described as "population balance and the sustainable distribution of our population in a way that works with the characteristics of our places and local ambitions for change." [82] (p. 66). The place-based ethos is evident in many of the intermediate outcomes that are described in the strategy document. For example, the activities of the three regional development agencies (Highlands and Islands Enterprise, South of Scotland Enterprise, and Scottish Enterprise), together with pilot initiatives in the Central Belt, are commended for exploring "a more regionally focused place-based model for economic

development" [82] (p. 77). A renewed housing and planning strategy supporting "self-build" housing is anticipated to contribute to a more balanced distribution of population. A pilot action for "community work hubs" to sustain the potential of homeworking and a focus on decentralisation of "anchor institutions" (e.g., public sector agencies) that should pay particular attention to private sector opportunities in associated local supply chains are both intended to strengthen local economies beyond the Central Belt.

Close ties exist, especially at the local level, between the public sector and the third sector, with voluntary and charitable organisations representing the core actors, who are often contracted to provide delivery of community services. Often this means that it is the third sector where innovation and local strategic planning take place, albeit within limits set by "arms-length" coordination of national strategy and local public sector commissioning. Considerations of community resilience [84] are thus pivotal to this strategy and might gain in relevance with further implementation. Community empowerment is a long-term concern of Scottish rural policy, although the assessment of the effectiveness of interventions is mixed [85].

The authors of the strategy claim [82] (p. 74) that the goal is mitigation, rather than adaptation: "Our focus in this programme is less on dealing with the impact of population change but rather focusing on the actions that need to be put in place to shift that change. Ensuring that Scotland's population is more balanced across the country means exploring the significant structural changes that are needed to support attraction and retention in those areas that are losing people and thereby reduce the pressure on areas dealing with a significant growth in population." However, a review of the policy actions cited (both existing and planned) shows a complex (hard to disentangle) combination of mitigation and adaptation. Prior to the recent increased interest at a national level, one gets the impression that the most common response was involuntary adaptation at a local level. Local governance (councils) has limited capacity and "mandate" to engage in mitigation strategies (such as place making and employment creation initiatives to attract migrants, or support for integration), although there have been some interesting pilot initiatives and there is currently a plan to establish "repopulation zones" under discussion [86]. Thus, in terms of the generic approaches in Table 1, the Scottish strategy is a complex hybrid of all four, in which moderate aspirations for mitigation at the national level are combined with smart adaptation in many local policy decisions.

*3.5. France—Mission Document for Equal Chances of Rural Areas (2020)*

With a large number of small villages and communities, France is particularly affected by spatial concentration processes that imply population decline in substantial pockets of its rural areas. Peripherality, and peripheralisation processes, are important long-term issues and recurrent topics, high on the spatial development agenda [87]. In the peripheralisation narrative "those who remain (in the remote regions of the country) are those who don't dispose of the resources for modern mobility. They also show a particular inclination to remain in their mode de vie, even if their world of life is widely regarded as obsolete" [88]. The growing sense of being ignored experienced by remote rural communities prompted an inter-ministerial committee to elaborate a new "Rural Agenda" [89], comprising a series of cross-sectoral instruments intended to mitigate adverse population trends.

A fundamental objective of regional policy, to achieve equality of chances for rural areas and small towns in France, was addressed in a dedicated mission statement [90], with the emotive sub-title "Restoring the promise of the Republic". This strategic document is conceived as part of the rural agenda, focusing primarily on the rural world, and proposing a bundle of activities and policy tools to address young people in rural regions. It is the result of numerous surveys, encounters, and workshops with local people, seeking to understand the specific concerns and required policy shifts to enhance the attractiveness of regions in population decline. The diagnosis emphasises the recursive human (and social) capital effects of selective out-migration in remote rural areas. These are seen not only in terms of the depletion of the skills base and the "draining-out" of entrepreneurial and

innovative young people, but also in the depression of morale and level of ambition of those who are left behind. Thus, the dominant generic policy approach appears to be one of relocalisation.

In spatial planning terms, the mission statement represents a switch from a dichotomous (centre-periphery) demographic development model to a vision of a more balanced spatial dynamic with a positive migration balance in the south-west and a negative one in the north-east of France. However, it is also fair to say that spatial development within rural France is, as elsewhere, highly localised, and the above generalisation could be judged to be simplistic. Aspects of the current discussion on the role of different administrative levels in a context of ongoing discourse on regionalisation (and localisation) are therefore crucial [91]. It is too early to assess whether the ambitions expressed in the mission document and the Rural Agenda will have an impact on local people's perspectives, raising their level of ambition, or triggering a process of activation. However, a wide range of activities, favouring an indirect mitigation approach, is outlined and specified as appropriate for the different types of rural regions, and inspiring local initiatives are reported as good practice examples.

## 4. Discussion

We began this paper by speculating that within the rural development discourse, the "back to basics" revival of interest in the elemental issue of demographic change might be indicative of (or at least associated with) a shift in the broader rural development paradigm, away from market-driven economic growth processes and towards a recognition of "softer", more qualitative objectives in the realm of well-being. We hypothesised that recent policy reviews by the EU reflected and built upon a prior shift in the discourse taking place at the national level. This is not to underestimate, of course, the influence of the evolution of rural development concepts and principles in other international forums, notably the OECD [43,92]. In a globalised world, the shift has likely affected all developed economies, and it is no surprise that OECD thinking and that of the European Commission are evolving in similar directions. However, the starting point for this shift was likely where the political impetus was most direct and forceful, at the national level.

To set the context for exploring selected national population strategies as influential in the wider rural development paradigm shift, we briefly summarised the evolution of policy at the EU level, specified some basic concepts of shrinking, reviewed relevant academic research findings, and proposed a simplified ToC framework as an aid to comparative assessment.

### 4.1. Shifting Policy Perspectives at the National Level

We now summarise what we have learned from the reviewed national strategies in terms of features and perspectives and highlight what they have contributed to the evolution of wider rural development paradigms:

1.  All the strategies tacitly acknowledge the need for interventions to go beyond the narrow, immediate demographic processes of out-migration, ageing, low fertility, and high mortality. Their responses confront *complex shrinking processes*, which require integrated solutions, breaking down policy silos. Whilst this is, conceptually, a strength, pragmatic observers may point to the risk of dissipation of thinly spread policy resources, a lack of coherence, and weak leadership.

2.  Another common thread in the strategies is the acknowledgement of the *plurality of rural contexts* (as opposed to an urban–rural dichotomy). The simplistic binary differentiation of urban and rural, recognised as unhelpful in academic circles for more than half a century [93], lingers on in the policy discourse. The strategies reviewed are notable in the degree to which they acknowledge the heterogeneity of rural contexts, both as a justification for place-sensitivity, and as a potential asset to be mobilised against (complex) shrinking processes.

3. A third observation relates to perceptions of the challenge and *diagnostic analogies*—the "pictures" that are used to describe the shrinking process and its complex effects. Generally, these imply that the aspatial/context-blind application of neoliberal notions of competitiveness or growth based upon innovation and entrepreneurship are insufficient. Four generic analogies have been observed:

    - *Emptying and filling areas*. This implies that areas are like containers that have a finite population capacity and that either emptying or overfilling has social and economic consequences. This analogy is explicit in the Italian and Spanish strategies.
    - *Regional (im)balance*—This analogy is closely related to the previous one but focuses on relative rates of change rather than deviation from a fixed capacity. Thus, in Scotland, the concern is about a drift from west to east; in France, from NE to SE; and in Germany, from the "New" Laender towards former West Germany. According to this perspective, the problem is not so much about absolute numbers of inhabitants, abandonment or "desertification", but more about the costs associated with a widening mismatch between population, service provision, infrastructure, and housing stock.
    - *Human/social capital disempowerment*—These are both causes and consequences of long-term demographic decline. Effects include the weakening of the ambitions and innovation capacity of younger people—the dominant concept driving the strategy in France.
    - *Well-being and spatial (in)justice*—Recursive effects of demographic change on living conditions and spatial inequalities, evaluated against notions of spatial justice, are key concepts in the German strategy and are also evident in the French documents.

4. *Intermediate Outcomes*—these are the specific changes delivered by interventions that are assumed to contribute towards mitigation. The strategies all focus mainly on (partial) mitigation rather than adaptation. Intermediate outcomes are multifarious, often not strictly defined, and range from housing market adjustments "reactivating" the ambitions of young people, increasing residential attractiveness, and promoting well-being. Whilst neo-liberal language (realising potential, promoting innovation, and entrepreneurship) remains, the balance has shifted away from market-driven economic growth, and towards more qualitative societal and individual benefits.

5. All four of the *generic strategies* described in Table 1 (compensation, relocalisation, global reconnection, and smart shrinking) are evident in the national policies described above. However, there are hints of shifting prioritisation, either in response to external forces (such as globalisation) or (particularly relocalisation and smart shrinking) as a consequence of deliberate reorientation and reflecting a rising awareness of unconventional options.

*4.2. Challenges for EU Policy: The Need for Coherence, Long-Term Commitment, Sensitivity and Trust*

Despite the increased attention at the strategic level by the commission, as evidenced by the LTVRA and the other documents cited above, specific policy commitments remain limited. Practical action through CAP Rural Development Programmes is, as yet, not much in evidence. Separately, in the cohesion policy context, the need for multilevel cooperation in response to demographic decline in remote regions has been emphasised [94]. However, Slee [95] (p. 45) observes "a consistent emphasis on the interconnected nature of all aspects of local economies and communities, and recognition that an integrated perspective is most beneficial, but often absent in public administration and policy making".

The challenge of coherence is not simply a matter of governance (i.e., ensuring that interventions through the CAP, cohesion policy, and other relevant policies are synergistic), it also requires a conscious coexistence of different intervention logics (as exemplified by

the four generic approaches featured in Table 1) to ensure that they tackle the bundle of challenges in an integrated way. The ESPON programme organisers have graphically synthesised findings from various ESPON projects in a summary design for "long-term territorially sensitive policies for the diverse rural shrinking areas" [96] (p. 18). This is a very effective way to communicate what coherence looks like in practice. Thus, building on local assets and enhancing the diversification of regional activities should be supplemented by concerns for distributional aspects and equalisation policies, as well as a focus on making use of technological developments, a concern for service provision, and a comprehensive vision for securing the well-being of all inhabitants (Figure 1).

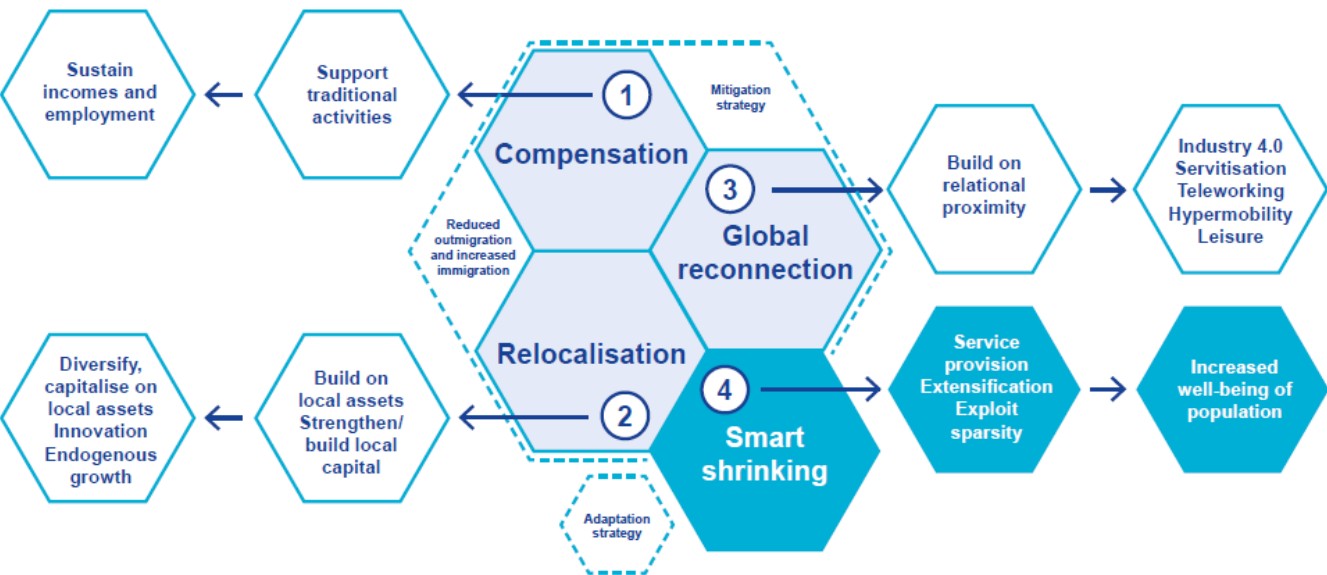

**Figure 1.** Redesigned framework for policy to address complex shrinking [96] (p. 21).

The four generic strategies proposed by the ESCAPE report [53] are placed in the centre of the diagram (hexagons 1–4). The distinction between mitigation (1–3) and adaptation strategies (4) is reflected in the colour of the hexagons. Surrounding these are examples of more specific "intermediate outcomes", which could act as strategic stepping stones through which mitigation and/or adaptation might be reached. It is essential that these different approaches interact and reinforce each other, to deliver maximum beneficial effects for rural regions.

As numerous studies suggest, the policy goals for altering socio-cultural and socio-economic trends in shrinking rural regions are not easy to realise [23,97]. The paradigm shift described above is a *sustained long-term policy commitment*. Institutional "lock-in" sectoral silos, and the paucity of trans-disciplinary research, all raise substantial barriers to successful implementation. There is an ever-present risk of overly optimistic views and exaggerated expectations.

The appropriate spatial scale/focus for interventions vary considerably between national contexts. On the one hand, in Spain, about 90% of the national territory is affected by shrinking. On the other hand, several countries, including Italy and Austria, focus on relatively small pockets of population decline [98]. Such intra-regional diversity brings with it added administrative complexity and poses particular questions in terms of governance, degree of detail in programming, and distribution of responsibility between administrative levels.

The introduction of novel approaches is often hampered by prevailing or "traditional" narratives. A good understanding of and sensitivity to local culture, perceptions, and "psychology" is necessary in order to avoid creating feelings of "spatial stigma". The current debate advocating a new perspective and more positive valorisation of the role of remote areas indicates the momentum for shifting the foundations of narratives in

these rural regions [99]. This discourse is not limited to the European context but has been taken up in many places across the world where people have been leaving for many years. Research needs to take account of the material and immaterial values of shrinking areas [100]. However, to corroborate this shift in public discourse, sufficient top-level support is required.

Building community-level trust involves the intensive exchange of perceptions between central and remote places to build common approaches and accommodate distinct views. Strategies that explore and acknowledge local views, inspirations, ideas, and innovative actions offer enormous potential to contribute to the evolving discourse. They support enhanced understanding of the potential of remote areas by people in distant places.

## 5. Conclusions

In this paper, we have explored the increasing interest in rural shrinking as a key theme in the long evolution of rural development policy [101]). Whilst there is a sense in which this is a return to one of its longest-established concerns, it can also be seen as totemic of some fundamental rethinking of underpinning assumptions. This advocates stepping off the treadmill in pursuit of economic growth and the fruits of urban development in favour of prioritising well-being and inclusion. Associated with this has been the (at least tacit) acknowledgement of several "elephants in the room", i.e.,

- Urbanisation, better links with cities and towns, and "spread effects" are not the only ways for rural areas to prosper;
- While primary sector activities play an important role in rural areas, rural policy should address the needs of the wider rural population, their economic activities, and service requirements;
- Rural areas are a heterogeneous group, with a variety of (economic, human, social, environmental) capital endowments, which can/should form the basis of new and unique place/community development paths;
- These new development approaches need to be "smart", both in the sense that they are place/context-appropriate and in the sense that they are responsive to changing technology and patterns of spatial behaviour, especially in terms of mobility, information and communication.

In this paper, we have acknowledged that such a reworking of rural development fundamentals has been initiated in a distributed, "bottom-up", incremental process in the context of national responses to rural population trends. The national examples described above reveal very distinct context-responsive features, but their central commitment is similar: to provide effective frameworks for local action, regional support, and national frameworks that enhance the capacity of rural areas to retain and attract population. This diversity of detail, combined with the emergence of common learning and principles, underlines the value of comparative analysis.

This is not to deny the importance of the parallel Brussels-led discourse and the various reincarnations of LEADER and the CLLD approach, which accelerate the process by providing helpful frameworks and facilitating the dissemination of innovative approaches. In a mid-term perspective, it will be important to explore what these shifts imply for existing EU rural development policy and future policy reforms. Such reflection has the potential to translate ongoing innovative discourses into lasting socio-political frameworks that would be conducive to more balanced territorial development.

**Supplementary Materials:** The following supporting information can be downloaded at: https://www.mdpi.com/article/10.3390/world3040053/s1.

**Author Contributions:** The conceptualisation, methodology, investigation, validation, literature review, writing-original draft preparation, and writing—review and editing has been carried out by both authors, T.D. and A.C. All authors have read and agreed to the published version of the manuscript.

**Funding:** This paper is based on the ESCAPE (European Shrinking Rural Areas: Challenges, Actions and Perspectives for Territorial Governance) project that has received funding from the ESPON 2020 programme. ESCAPE project was funded by ESPON EGTC, Service contract—EE/SO1/071/2019.

**Institutional Review Board Statement:** Not applicable.

**Informed Consent Statement:** Not applicable.

**Data Availability Statement:** Not Applicable.

**Conflicts of Interest:** The authors declare no conflict of interest.

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
