# Peer review of "European Rural Demographic Strategies: Foreshadowing Post-Lisbon Rural Development Policy?"

_world, doi:10.3390/world3040053_

Round 1

Reviewer 1 Report

The paper covers an important issue, as it shows patterns in thinking about dealing with depopulation of rural areas within European countries.  Generally, it is interesting and well-written. There are few minor suggestions to improve it.

Looking for patterns in each of the strategies and similarities between the countries is a good idea (section 3). I suggest strengthening a bit the reference to certain diagnostic narratives and intermediate outcomes (perhaps some comments in the section 3 referring to the Table 1).

In the case of Germany, Scotland and France I didn’t see a clear comment concerning the proportions of mitigation and adaptation strategies (such comments were written in the case of Spain and Italy). If making such comments is possible, it would make the text clearer.

A bit more concrete examples of activities and measures suggested in the country strategies that are not market-driven would be beneficial (either in the text or as a table in an appendix).

Section 4.2 “Challenges for EU policy: Coherence, long-term commitment, and trust” seems somehow blurred to me. Personally, I would shorten it and place it before the section 3, but I leave it to your decision.  

Author Response

Please find our comments attached.

Reviewer 2 Report

The study presents an interesting analysis of the changes in the policy approach to supporting rural areas. This study constitutes a strong and thought provoking statement in the discussions on the EU, national and regional policy aimed at supporting rural areas in coping with the current challenges of depopulation, aging and climate change. I greatly appreciate the fact that this study analysis policy changes and challenges from a longer perspective than it is typically done.

Author Response

Please find our comments to reviewer 2 attached.

Reviewer 3 Report

In the context of an increasing attention devoted to demographic trends, especially in rural Europe, the paper explores a series of national population strategies.

Authors conduct a comparative review of these strategies, identifying different nuances.

The paper is well structured and contextualized, and provides an interesting and valuable contribution to the debate on the issues at the heart of the discourse.

I recommend to go through hthe whole paper and take care of minor revisions (e.g. 2008 Financial Crisis not in capital letters; line 563, there is a bracket to be cancelled).

Author Response

Please find attached our answer to Reviewer 3 comments.
